# Nuclear-Mitochondrial Interactions

**DOI:** 10.3390/biom12030427

**Published:** 2022-03-10

**Authors:** Brittni R. Walker, Carlos T. Moraes

**Affiliations:** 1Neuroscience Program, University of Miami Miller School of Medicine, 1420 NW 9th Avenue, Rm. 229, Miami, FL 33136, USA; brw52@miami.edu; 2Department of Neurology, University of Miami Miller School of Medicine, 1420 NW 9th Avenue, Rm. 229, Miami, FL 33136, USA

**Keywords:** mitochondria, nucleus, retrograde signaling, MAMs, integrated stress response

## Abstract

Mitochondria, the cell’s major energy producers, also act as signaling hubs, interacting with other organelles both directly and indirectly. Despite having its own circular genome, the majority of mitochondrial proteins are encoded by nuclear DNA. To respond to changes in cell physiology, the mitochondria must send signals to the nucleus, which can, in turn, upregulate gene expression to alter metabolism or initiate a stress response. This is known as retrograde signaling. A variety of stimuli and pathways fall under the retrograde signaling umbrella. Mitochondrial dysfunction has already been shown to have severe implications for human health. Disruption of retrograde signaling, whether directly associated with mitochondrial dysfunction or cellular environmental changes, may also contribute to pathological deficits. In this review, we discuss known signaling pathways between the mitochondria and the nucleus, examine the possibility of direct contacts, and identify pathological consequences of an altered relationship.

## 1. Mitochondrial Form and Function

Mitochondria, a double membrane organelle evolving from an engulfed α-proteobacterium, is colloquially referred to as the powerhouse of the cell [1,2]. The mitochondrion consists of an outer membrane, an intermembrane space, the inner membrane, and the matrix. The outer mitochondrial membrane (OMM) contains porins, which allow for the relatively free diffusion of small molecules into the intermembrane space. The intermembrane space also contains proteins involved in bioenergetics and apoptosis. The inner membrane is highly impermeable and contains many transporters for mitochondrial proteins. The inner mitochondrial membrane (IMM) creates folds into the matrix, known as cristae, where the respiratory complexes are located [3]. Finally, the matrix is the innermost space where many other processes, such as the Krebs cycle, mtDNA replication, transcription, and mitochondrial protein synthesis, occur [4,5]. Independent of the nuclear genome, mitochondrial DNA (mtDNA) are double-stranded, circular molecules of approximately 16.5 kilobases, encoding for 13 proteins of the oxidative phosphorylation (OXPHOS) complexes, 22 tRNAs, and 2 rRNAs in mammals [6,7]. It is a multicopy genome, meaning each cell may hold hundreds to thousands of copies of mtDNA [8]. Although present in the matrix, the mtDNA associates with several proteins, forming nucleoids, which are present in the inner aspect of the inner membrane [9].

Mitochondria exist both as individual organelles and highly connected networks. The network is constantly transformed through fusion and fission, two major aspects of mitochondrial dynamics, which aids in maintaining both mitochondrial and cellular health [10]. Fusion is the process of two mitochondrial segments merging together; the outer membranes must fuse together first, then the inner membranes before an exchange of information (mtDNA, proteins, etc.) can occur [10]. As cellular cues influence fusion events, it is thought that fusion acts to marry the state of the cell with mitochondrial function [10]. Mitofusin 1 (Mfn1), Mitofusin 2 (Mfn2), and OPA1 are the three major fusion proteins, with the first two present at the OMM and Opa1 at the IMM. Knockout of these proteins is embryonic lethal, and mutations are associated with neurodegenerative diseases [11,12,13]. Fission is a division event that occurs at the endoplasmic reticulum (ER)—mitochondria contact sites and results in one or more daughter mitochondria [10]. Knockout of fission-associated proteins leads to elongated mitochondria, while overexpression leads to fragmented mitochondria [14]. Dynamin-related protein 1 (DRP1) coordinates mitochondrial fission by “strangulating” rings formed around the OMM [15]. Dysregulation of fission events has detrimental effects, including congenital microcephaly, and may play a role in Huntington’s Disease and Alzheimer’s Disease [16].

Mitochondria earned their nickname, Powerhouse of the Cell, from their major role in energy production [2]. Oxidative phosphorylation is the process in which the mitochondria convert substrates into ATP. As mentioned above, the respiratory chain is embedded in the crista and consists of four large protein complexes, ATP synthase, and two mobile electron transporters. In addition to ATP, reactive oxygen species (ROS) are produced during this process, which acts as signaling molecules. In addition to significant contributions to bioenergetics, mitochondria also play a role in several other metabolic pathways, such as phospholipids biosynthesis, Ca^2+^ regulation and signaling, and cellular stress response [17,18,19].

### Mitochondrial Transport and Distribution

Mitochondria utilize the cytoskeletal network to move throughout the cell [20]. Microtubules are used for long-range transport, while the actin network is used for short-range movements [20,21]. Via the adaptor complex Miro, which anchors to the outer mitochondria membrane, and Milton, which acts as an adaptor between Miro and the motor protein, the mitochondria can undergo long-range movements [21]. For microtubule-mediated transport, the kinesin superfamily proteins are the anterograde (towards the cell periphery) motors, whereas dynein drives movement in a retrograde function (towards the nucleus) [21,22]. Previous literature has shown that the knockout of the kinesin heavy chain in mice is embryonic lethal, and further analysis found kinesin disruption to be correlated with perinuclear clustering of mitochondria [22,23]. Myo19 has been identified as a motor protein for the actin network [24]. Myo19 requires the Miro proteins for stabilization and recruitment to the mitochondrial membrane [25]. Knockout of the Miro proteins, and the consequent degradation of Myo19, results in perinuclear clustering and asymmetric segregation of mitochondria during mitosis [25].

In general, mitochondrial trafficking allows for the distribution of mitochondria to respond to local demands for ATP and Ca^2+^ buffering [21]. The motility of mitochondria is influenced by cytosolic Ca^2+^, ROS, AMPK signaling, and other factors [21,26,27,28,29]. Additionally, there is cell-specific significance for mitochondrial transport. For example, neuronal development, axon regeneration, and axon branching are influenced by the spatial and temporal regulation of mitochondria [30,31,32]. Redistribution of mitochondria within lymphocytes aids cell migration and polarity during an immune response [33]. Most importantly, mitochondrial trafficking allows for proper distribution during embryonic development and mitosis [34,35].

Mitochondrial distribution, while heterogenous, has some consistent patterns across various cell lines. Collins et al. performed a study to visualize the mitochondrial network, and in all cells studied, mitochondria were distributed throughout the cytosol but had a higher density in the perinuclear region [36]. However, there is cell-specific organization as well. For example, neurons tend to have larger populations of mitochondria at synaptic sites to provide energy for neurotransmission [37]. In mature cardiomyocytes, mitochondria have three distributions: long rows between bundles of myofilaments, under the sarcolemma, and perinuclear clusters [38]. Collins et al. concluded that the heterogeneity in mitochondrial morphology aids the organelle in functioning independently with respect to mitochondrial membrane potential, Ca^2+^ sequestering, and permeability transition pore activation [36]. The combination of transportation and mitochondrial dynamics allows for the mitochondria to alter the extent of their connectivity [36].

To maintain optimum distribution, mitochondria must have an anchoring system. Several anchors have been identified in yeast, but this system is less clear in mammalian cells [39]. In addition to the cytoskeleton, it is likely that mitochondria use tethering between other cellular membranes as a sort of anchor [39,40,41]. Although syntaphilin was previously identified as a neuron-specific mitochondrial docking protein, Caino et al. observed syntaphilin mRNA expression in non-neuronal tissues and cell types [42,43]. Depletion of this protein also has tumor-enhancing effects, allowing for mitochondria to accumulate at the leading edge of the cell and supporting tumor cell migration [43,44].

## 2. Physical Interactions of Mitochondria with the Nucleus

Mitochondria-associated membranes (MAM) are physical associations between the mitochondria and other organelles, such as ER, lysosomes, and peroxisomes. The existence of these contact sites suggests functions in addition to organelle-specific tasks. For example, it has been found that the domain formed between the endoplasmic reticulum (ER) and the mitochondria—the best characterized MAM—functions in regulating lipid synthesis, Ca^2+^ signaling, controlling mitochondrial biogenesis, and intracellular trafficking [27]. Through studying ER-mitochondrial contacts, it has been observed that the two organelles can tether together through stable interactions between apposing membrane proteins [27]. While tethers between the mitochondria and the ER have been well established in yeast (ERMES), these connections are less clear in mammals. A number of tethering complexes and proteins have been proposed, such as IP3R/Grp75/VDAC, PTPIP51/VAPB, and Mfn2; however, no one protein appears to be sufficient for maintaining MAM structure and integrity [45]. As much as 20% of the mitochondria surface is juxtaposed to the ER, with approximately 10–30 nm between the organelles [46]. Ion transporters and biosynthetic enzymes are prominent components of MAMs [46]. Another major feature of MAMs is the lipid raft-like domains, enriched in cholesterol and gangliosides [47]. Lipid composition influences apoptosis, autophagy, and as well as mitochondrial dynamics and morphology [45,46]. Alterations in MAM signaling and contacts have been associated with various pathologies, such as cancer, diabetes, and neurodegenerative diseases [28].

The nucleus has two enveloping membranes, an inner (INM) and an outer nuclear membrane (ONM), joined periodically by nuclear pore complexes [29]. The ONM is contiguous with both rough and smooth endoplasmic reticulum (ER) [29]. As the ONM and ER are contiguous, and ER-mitochondrial contacts are well established, it seems likely that these two membranes may also come into close contact. However, stable MAM structures between mitochondria and nucleus have not been described.

Early studies of the mitochondria and their relation to the nucleus hinged on the “mystery” of mitochondrial origin in the cell—where did new mitochondria come from? There were several speculations, including de novo formation and binary fission [48,49], but a common belief was mitochondrial were formed and extruded from the nucleus [50,51]. Electron microscopy showed a close association of the mitochondria with the nucleus in various cell types [50,51,52,53]. Mota described an accumulation of mitochondria within invaginations of the nuclei of aerial roots of *Chlorophytum capense* [53]. The contacts have even been described as the mitochondrial and nuclear membranes being contiguous [53,54]. A study by Frederic showed that these interactions increased with the addition of 2,4-dinitrophenol, an OXPHOS uncoupler [55].

Prachař observed mitochondria in close proximity of the nuclear envelope in L1210 mouse leukemia cells, also noting a fusion of the outer membranes [56]. The fusion occurred at a much higher incident rate in the rapidly growing L1210 cells, in comparison to others, possibly due to increased metabolic activity [56]. As mitochondria can be seen perinuclearly in almost of metabolically active cells, Prachař suggested the contacts between the nucleus and the mitochondria could act as an energy reservoir for mRNA and protein transport [56]. Dzeja et al. also proposes energetic communication as a major function of nuclear-mitochondrial contacts [57]. Through laser confocal microscopy, they observed mitochondria clustered around the nucleus, although structures in the perinuclear space hindered direct contact, and hypothesized this proximity is required due to the high-energy demands of the nucleus [57]. ATP is required for nuclear transport, and more specifically, inhibition of OXPHOS abolishes transport while inhibitors of glycolysis decreased ATP production but did not abolish transport [57].

The proximity between these organelles may also function to accelerate retrograde responses, stimulating mitogenesis or mitophagy. In a study by Al-Mehdi et al., hypoxia triggers the perinuclear localization of mitochondria in pulmonary artery endothelial cells (PAEC) [23]. Consequently, reactive oxygen species (ROS) then accumulate in the perinuclear and nuclear regions, introducing an oxidative base modification in hypoxia response elements of hypoxia-inducible promotors, important for transcriptional activation [23]. In hypoxic conditions, hypoxia-induced factor 1α (HIF-1α) induces transcription of hypoxia up-regulated mitochondrial movement regulator (HUMMR) [58]. Interestingly, Li et al. found perinuclear clustering following transfection with HUMMR in mouse astrocytes; however, it was also observed that HUMMR functions to increase anterograde movement while decreasing retrograde mitochondrial movement [58].

More recently, Desai et al. described nuclear-mitochondrial contact sites that aid in a pro-survival retrograde response in MDA cells [59]. In their study, they identified TSPO, an OMM-localized protein, as a key protein in a scaffolding complex between the two organelles [59]. Upon induction of mitochondrial stress, mitochondria redistribute to the perinuclear region and deliver cholesterol to the nuclear envelope [59]. The contact site increases nuclear exposure of ROS and nuclear stabilization of pro-survival transcription factors [59]. Additionally, it was suggested that TSPO deficiency triggers a retrograde response via disruption of the mitochondrial membrane potential (∆Ψm) [60]. The decrease in ∆Ψm leads to a dysregulation of calcium homeostasis, reduced respiratory function, and altered transcriptome [60].

## 3. Functional Interactions between the Mitochondria and the Nucleus

### 3.1. Nuclear Control

When mitochondria evolved from endosymbionts to organelles, they experienced a massive genome reduction [1,61,62]. The majority of the ancient mitochondria’s genome was transferred to the host’s nucleus and integrated into the eukaryotic genome. Nuclear-encoded mitochondrial genes include the outer membrane and intermembrane space proteins, as well as most inner membrane and matrix proteins [63]. The proteins are initially synthesized on free cytosolic ribosomes [64]. Many precursor proteins contain an N-terminal mitochondrial targeting sequence (N-MTS) which is recognized by receptors on the mitochondrial surface and eventually cleaved [63]. Other precursor proteins contain an internal targeting sequence that will not be cleaved [63]. Although there are multiple import pathways, the canonical pathway involves initial import through the translocase of the outer membrane (TOM) complex [64]. From there, the precursor proteins will be processed by the TIM22 or TIM23 complex resulting in import to the matrix, integration into the outer and inner membrane, or released into the intermembrane space [64].

All transcription factors involved in mitochondrial gene expression are encoded by the nucleus, as well as the major transcriptional co-activators [65]. Nuclear respiratory factors 1 and 2 (NRF-1/NRF-2) are nuclear-encoded transcription factors that activate nuclear-encoded genes coding for mitochondrial proteins [65]. NRF-1 acts on the majority of genes required for mitochondrial respiratory function, plus genes encoding components of the heme biosynthetic pathway and the protein import and assembly complex [65]. NRF-2 activates cytochrome oxidase subunit IV and three of the succinate dehydrogenase (Complex II) subunits [65]. Most important NRF-1 and -2 activate mitochondrial transcription factor A (TFAM) [65,66]. In addition to its role in transcription initiation, TFAM contributes to the stabilization and maintenance of mtDNA [65]. Peroxisome proliferator-activated receptor gamma coactivator 1α (PGC-1α) is the major transcriptional coactivator for mitochondrial biogenesis [65]. PGC-1α interacts with NRF-1 and -2, as well as estrogen-related receptor alpha and several other tissue-specific transcription factors to activate transcription of nuclear genes encoding mitochondrial proteins, including TFAM [65], mitochondrial RNA polymerase (POLRMT), the initiation factor TFB2M, and transcription termination factors mTERFs [65]. The nucleus also controls replication, maintenance, and segregation of mtDNA [67,68]. Table 1 summarizes the role of nuclear factors in responding to mitochondrial signals.

### 3.2. Retrograde Signaling

As described above, over 95% of mitochondrial proteins are coded by nuclear DNA. Therefore, alterations and adaptations in mitochondrial function are heavily dependent on the nucleus responding to signals originating at the mitochondria, a process known as “retrograde signaling” [90]. Retrograde signaling can extend lifespan by adapting to metabolic needs and eliminating dysfunctional organelles [91]. The retrograde response may be stimulated by the ATP/ADP ratio, disruption of the mitochondrial membrane potential, reactive oxygen species (ROS), or general cellular stress [92,93].

The retrograde signaling response was first discovered and characterized in yeast [94,95]. In yeast, Rtg1-3 has been identified as a direct mediator of the retrograde response [96,97]. Upon assembly, the Rtg complex translocates from the cytoplasm to the nucleus [98]. The RTG pathway compensates for mitochondrial dysfunction by upregulating citric acid cycle genes and, therefore, citric acid cycle activity [75]. Studies showed that ATP concentration is a major trigger of the retrograde response in yeast [99]. Inhibition of the TOR pathway, as well as activation of the SIRT2 pathway, have been shown to activate retrograde signaling [75,100].

In *C. elegans* and *Drosophila*, the retrograde response is primarily carried through via the mitochondrial unfolded protein response, which will be discussed more below [92].

While retrograde signaling pathways have been elucidated in yeast, many of the identified proteins do not have a mammalian homolog, and there is still controversy whether there is a bona fide mitochondrial UPR or how to define the mitochondrial stress response in mammalian cells [101,102]. Here we will discuss signaling molecules and pathways that fall under the category of retrograde signaling. Table 2 contains a summary of the major signaling molecules involved in canonical mito-nuclear communication.

#### 3.2.1. Calcium

In partnership with the ER, the mitochondria regulate Ca^2+^ homeostasis. As the ER acts as the largest store of cellular calcium, microdomains of high calcium concentration can form at ER-MAMs. Mitochondria can uptake calcium through the OMM via VDAC and use the MCU (mitochondrial calcium uniporter) to bring calcium into the matrix. Mitochondrial buffering of calcium regulates the activity of calcium channels in a negative feedback loop. Within the mitochondria, calcium concentration greatly affects mitochondrial function—several dehydrogenases in the matrix are sensitive to Ca^2+^, therefore influencing ATP synthesis via NADH availability and electron flow [133]. High concentrations of Ca^2+^ can stimulate the opening of the permeability transition pore (PTP) and induce apoptosis or necrosis, while low concentrations may stimulate pro-survival autophagy due to decrease ATP concentrations [133].

Calcium acts as a second messenger in many signaling pathways, its spatial and temporal waves regulate the activation of transcription factors and, therefore, gene expression [103,106]. Calcium was first observed as a retrograde signaling molecule in a study by Biswas et al. [106]. Through a combination of mtDNA depletion and metabolic inhibitors, they were able to establish that disruption of the mitochondrial membrane potential and altered ATP synthesis leads to Ca^2+^ efflux in C2C12 muscle cell lines [106]. They observed decreased NF-κB activity, as well as an increase in JNK-dependent ATF2 and calcineurin-dependent NFAT [106]. As a note, the NF-κB pathway is thought to evolve from yeast retrograde signaling [110]. In response to mitochondrial respiratory stress, NF-κB translocates to the nucleus to activate transcription of target genes for Ca^2+^ homeostasis and glucose metabolism [104,110]. Heterogenous nuclear ribonucleoprotein A2 acts as a co-activator by associating with the enhanceosome and acetylating target promoters [109,111]. Additionally, they saw increased expression of ryanodine receptor-1 (RyR-1) and subsequent Ca^2+^ release [106]. More recent research supports this model, showing mitochondrial dysfunction upregulated RyR-1 activity as well as decreased levels of its regulator, FKBP12, resulting in intracellular Ca^2+^ leak and calcineurin-dependent retrograde signaling [108]. Ultimately, in cells experiencing mitochondrial dysfunction, increased cytoplasmic Ca^2+^ leads to alteration in the activity of pro-inflammatory and cell proliferation transcription factors and even expression of anti-apoptotic markers [105,106,107]. Figure 1 illustrates the role of several molecules in retrograde signaling, discussed in this review.

#### 3.2.2. Free Radicals

Free radicals are any molecular species capable of independent existence that contains an unpaired electron in an atomic orbital. Many are unstable and highly reactive, leading to the early belief that all free radicals are exclusively damaging agents. While excessive levels of free radicals can be damaging, especially to DNA, it has also been shown that they are essential for numerous signaling pathways, including retrograde signaling [112,114,134]. Reactive oxygen species (ROS) include superoxide (O_2_^−^), hydrogen peroxide (H_2_O_2_), and the hydroxyl radical (OH•) [135]. Mitochondrial produced ROS (mtROS) primarily forms at complex I and III of the respiratory chain [134]. It has roles in apoptosis, activation of transcription factors, cell differentiation, and aging [112]. ROS activates retrograde signaling pathways in a dose-dependent manner—lower levels may induce the Ca^2+^/Cn pathway while higher levels may reflect hypoxic conditions [136].

ROS primarily activates stress retrograde pathways; however, mtROS can act as a direct retrograde molecule as well, specifically in hypoxic conditions. During hypoxia, mtROS stabilizes hypoxic induction factors, specifically HIF-1α, which allows for the transcriptional response to hypoxia, the dampening of ROS production, and ultimately increases replicative life span [112,113]. As previously described above, Al-Mehdi et al. observed that perinuclear clustering of mitochondria during hypoxia enhanced the ability of mtROS to accumulate near the nucleus and induce oxidative base modifications of the VEGF promoter [23]. These modifications are important for the hypoxic transcriptional response [23].

Although we primarily discuss pro-survival pathways in this review, it is important to note that some retrograde signaling pathways lead to apoptosis; such as ROS-induced JNK (c-Jun N-terminal kinase) signaling. JNK is a member of the mitogen-activated protein kinase superfamily (MAPKs) [137]. Depending on the stimuli and cell type, JNK can promote apoptosis or cell survival [138]. ROS activates JNK through upstream kinases, including ASK1 (apoptosis signaling regulated kinase 1) and Src [139]. Prolonged activation of JNK can stimulate a second generation of mitochondrial superoxide and promote pro-apoptotic activation through interactions with the Bcl-2 family [139,140]. In contrast, transient activation of JNK may promote pro-survival signaling, potentially enhanced by other survival signaling pathways, such as NF-κB [137,141,142].

Nitric oxide (NO) is a freely diffusible gas synthesized from L-arginine and O_2_ by NO synthase in a Ca^2+^-dependent manner [117]. The existence of mitochondrial NO synthase is highly debated; however, NO has been detected within the mitochondria and shown to regulate mitochondrial function [115]. In addition to acting on the respiratory chain complexes, NO has been shown to be a player in mitogenesis [114,115,117]. In a cGMP-dependent manner, NO increases expression of PGC-1α, which subsequently increases expression of transcription factors NRF-1 and TFAM [115,117]. Following exposure to NO, Nisoli et al. observed an increase in mtDNA content and functionally active mitochondria in mammalian cells [116].

#### 3.2.3. Metabolic Sensors

##### AMPK Pathway

AMP-activated protein kinase (AMPK) is a metabolic sensor of the AMP/ATP ratio [118,119]. It is activated by cellular stress, fasting, and exercise (increase in AMP) and acts as a switch for catabolic pathways to generate ATP [119]. At the same time, it inhibits ATP-dependent biosynthetic pathways to reserve cellular ATP [119]. In recent years, scientists have been able to resolve a pool of AMPK with mitochondrial localization [120]. Inhibition of mitochondrial AMPK activity was sufficient to trigger cytosolic ATP increase [120]. AMPK has several effects on mitochondrial function, including mitogenesis, mitophagy, and regulation of mitochondrial dynamics [121,122,123,124].

##### mTOR Pathway

Mammalian target of rapamycin (mTOR) is a serine/threonine kinase that regulates anabolic processes in response to growth factors, energy status, and oxygen levels [143]. mTOR complex 1 (mTORC1) is a central signaling complex very sensitive to rapamycin that stimulates protein synthesis and other anabolic processes, including mitogenesis and mitochondrial activity [143]. mTORC1 controls these functions through phosphorylation of 4E-BP, eIF4E-binding proteins, allowing for the assembly of the translation initiation complex [144]. It can also modulate energy metabolism through stimulation of PGC-1α, HIF-1 α, and SREBP1/2 [144]. Finally, mTORC1 is thought to play a role in the Integrated Stress Response by remodeling one-carbon metabolism [145].

##### Sirtuins

Three sirtuins (SIRTs), SIRT3, 4, and 5, are localized in mitochondria and respond to low mitochondrial membrane potential [146]. SIRT3 and SIRT5 are NAD(+)-dependent deacetylases, removing acetyl groups from acetyllysine-modified proteins. SIRT4 transfers the ADP-ribose group from NAD(+) to acceptor proteins. With adenine nucleotide translocator 2 (ANT2), SIRT4 regulates the coupling efficiency of mitochondrial respiration. SIRT4 acts in a feedback loop with ANT2-AMPK-PGC1α to regulate mitochondrial mass and transcription of OXPHOS genes [147]. In this loop, overexpression of SIRT4 leads to an increase in ATP, which decreases phosphorylated AMPK [147]. Consequently, this decreases the expression of fatty acid oxidation genes and PGC1α activity [147]. Additionally, SIRT4 positively regulates mTORC1 signaling [148]. As SIRT4 is active in a “fed” state, it mediates “glutamine sparing” via inhibition of glutamate dehydrogenase [148]. This allows mTORC1 to be active and induce anabolic pathways [148].

##### FOXO Factors

Forkhead box O (FOXO) transcription factors are thought to have a role in retrograde signaling; however, it is not yet determined whether this role is direct, indirect, or both [149].

FOXO transcription factors are inhibited by insulin and growth factor signaling and regulated by phosphorylation and post-translational modifications [149]. FOXO activity and function are also regulated by ROS, AMP, NAD+, and Acetyl-CoA, further implicating it in retrograde signaling pathways [149,150]. In states of high cellular ROS, FOXO3 is able to upregulate mtROS scavengers while reducing mitochondrial function to prevent ROS generation [149]. FOXO3 also mediated PINK1 expression and can therefore regulate mitochondrial remodeling and mitophagy [149,151].

#### 3.2.4. Mitochondrial Derived Peptides

Mitochondrial derived peptides (MDPs) are peptides encoded within small open reading frames in the mtDNA. Three types have been reported, MOTS-c (mitochondrial open-reading frame of the twelve S rRNA-c), Humanin, and SHLPs (small humanin-like peptides), and they were proposed to play roles in metabolism, aging, and cell survival [152]. MOTS-c is a 16-amino acid peptide encoded with the mitochondrial 12S ribosomal RNA [153]. MOTS-c regulates cellular metabolism in an AMPK-dependent manner [153]. Under metabolic stress conditions, MOTS-c translocates from the mitochondria to the nucleus. MOTS-c binds to Antioxidant Response Elements (ARE) on the nuclear DNA and interacts with NRF2 to activate transcription of stress response genes [154]. Humanin is a 24-amino acid peptide encoded with the mitochondrial 16S ribosomal RNA [155]. Humanin can bind cell surface receptors that activate signaling pathways for cell proliferation and survival, as well as block apoptosis, decrease inflammation, and reduce oxidative stress in various aging models [155]. MOTS-c and humanin increase during senescence and increase senescence-associated secretory phenotypes. Furthermore, this increases respiration via fatty acid oxidation [155]. SHLPs are six, 20–38-amino acid peptides encoded within the 16S ribosomal RNA [152]. Although SHLPs have been studied less, it has been shown that they have organ-specific expression [152]. SHLP 2 and 3 appear to have a cytoprotective role, while SHLP 6 increased apoptosis [152].

Although there have been several studies studying the response to humanin, the origin of the peptide is still controversial. MtDNA gene sequences are present in the nucleus and several species, including macaques, showed that the peptide would likely not be produced in the mitochondria, as it lacks the initiator methionine [156].

#### 3.2.5. mtDNA

The mtDNA also acts as a signaling molecule. It has previously been shown that cytoplasmic mtDNA and mtRNA activate the cGAS-STING-TBK1 pathway, stimulating an antiviral immune response [157,158]. Wu et al. developed a *Tfam^+/−^* MEF cell line that, due to its reduced expression of TFAM, exhibited elongated mitochondria, enlarged nucleoids, and increased basal release of mtDNA [159]. When exposed to mitochondrial stressors, *Tfam^+/−^* MEFs were more resistant to cell death compared to wild-type cells [159]. Through DNA-damage response induction/repair studies, it was found *Tfam^+/−^* MEFs had faster nDNA repair kinetics, an effect not seen in ρ° cells, due to their lack of mtDNA [159]. Upregulation of PARP9, an Interferon Stimulated Gene, played a major role in the DNA repair pathway [159].

MtDNA stress signaling may also have negative effects, as seen by Hamalaninen et al. Mutation of the mtDNA replicase, Polg, not only increases mtDNA mutations but also stalled the cell cycle and increases the frequency of double-stranded DNA breaks [160]. Although nucleotide pools increase up to tenfold in preparation for the S phase, total cellular dNTP pools were significantly diminished in mutator iPSCs [160]. The dNTP pools, especially dTTP and dATP, were preferentially sequestered into the mitochondria for mtDNA maintenance [160]. As the Polg mutation induced a stress-related phenotype, the frequency of mtDNA replication was increased, driving the demand for nucleotides up and destabilizing the nuclear genome [160].

#### 3.2.6. Non-Coding RNAs

Non-coding RNAs (ncRNAs) are RNAs of varying lengths that do not code for a protein. These are not RNA turnover products, but rather ncRNAs are generated by ribonucleases [161]. ncRNAs have cell-specific expression patterns and bidirectional signaling [162]. Although the majority of ncRNAs are encoded by the vast nuclear genome, the mtDNA encodes mitochondrial ncRNAs as well [163]. Both nuclear and mitochondrial ncRNAs were reported to localize in the mitochondria [163]. More recently, mitochondrial ncRNAs have been suspected of playing a role in retrograde signaling. Nonetheless, this is also a controversial topic, as the possible mechanisms associated with their production and function are not known [164].

mito-ncR-805 is a non-coding RNA that maps to the light-strand promoter D-loop regulatory region of mtDNA [165]. It is a 70 bp transcript that has a granular form, concentrated in the perinuclear region [165]. In alveolar epithelial Type-I (AETII) cells, upon cigarette smoke exposure, mito-ncR-805 was suggested to localize in the nucleus, where it regulates a subset of nuclear-encoded mitochondrial genes, particularly TCA cycle enzymes and respiratory chain complex subunits [165]. Again, it is unclear whether this RNA originates from nuclear pseudogenes.

microRNAs (miRNAs) are the most studied ncRNAs. They are around 22 nucleotides and regulate the translation of mRNAs into proteins [166]. It has already been suggested that miRNAs can regulate anterograde signaling, for example, through modulating COX1 expression and mitochondrial morphology, but less has been published about miRNAs in the retrograde response [167]. Although transcribed by the nuclear DNA, miR-663 acts as a regulator of retrograde signaling [167]. miR-663 regulates OXPHOS complex activity by controlling the expression of nuclear-encoded OXPHOS subunits and assembly factors, as well as stabilizing supercomplexes [167]. In oxidative stress conditions, increased ROS levels and hypermethylation of the miR-663 promoter resulted in decreased expression of miR-663 [167]. This, in turn, reduces OXPHOS gene expression and ultimately OXPHOS capacity [167].

Other interesting ncRNAs involved in mito-nuclear communication include the Telomerase RNA TERC, SncmtRNA, and tRFs. Telomerase RNA TERC was reported to be imported into the mitochondria, then processed into TERC-53 before being exported back into the cytosol [168]. The cytosolic levels of TERC-53 reflect the mitochondrial state—mitochondrial dysfunction leads to the accumulation of TERC-53 [168]. TERC-53 regulates GAPDH translocation into the nucleus and appears to play a role in cellular senescence [169].

Sense mitochondrial encoded ncRNA (sncmtRNA) is a 2374 bp transcript that contains a loop forming inverted repeat, jointed at the 5′ terminus of the 16S mitochondrial ribosomal RNA [170]. It is expressed in proliferating cells and tumor cells, localizing in the mitochondria, the cytoplasm, and the nucleus [171]. Within the nucleus, Landerer et al. observed an association of sncmtRNA with heterochromatin, where it may play an epigenetic role on cell cycle progression [171]. tRNA fragments (tRFs) are nucleotide fragments produced from nuclear- and mitochondrial-encoded tRNA loci [172]. There are five types of tRFs—5′ fragments, 3′ fragments, 5′ halves, 3′ halves, and internal tRFs [172]. Like most ncRNAs, tRFs are very specific to many conditions, including cell type, tissue type, disease, and an individual’s characteristics [172]. tRFs have been shown to influence apoptosis, translation, and the viral response [173]. Although the exact role of mitochondrial tRFs has yet to be elucidated, their detection in various pathophysiological conditions suggests a role in the cellular stress response [172,173,174,175,176]. Definite proof that these RNAs migrate from the mitochondria to the nucleus is not available, as the presence of hundreds of mitochondrial DNA pseudogenes in the nucleus makes this assumption controversial.

#### 3.2.7. Integrated Stress Response

The Integrated Stress Response (ISR) is a signaling network that helps the cells adapt to environmental and pathological conditions [177]. Triggers of the ISR include the unfolded protein response, nutrient deprivation, viral infection, oxidative stress, and mitochondrial dysfunction [177,178]. The stresses are sensed by four kinases, HRI, GCN2, PERK, and PKR, that phosphorylate eukaryotic translation initiation factor (eIF2α) [177,179]. This results in the reduction in protein synthesis while also stimulating the translation of ISR-specific mRNAs that inhibit transcription initiation [179]. Below we will briefly discuss some of these triggers and the major players in their pathways in relation to the mitochondria.

##### UPR^mt^

The mitochondrial unfolded protein response (UPR^mt^) is a protective transcriptional response triggered by mitochondrial proteotoxic stresses, including the accumulation of unfolded or misfolded proteins and mitochondrial dysfunction [180]. This pathway was elucidated in *C. elegans*, but it probably differs somehow in mammalian cells [181]. The dual localizing transcription factor, ATFS-1 is imported into the mitochondria where its mitochondrial targeting sequence (MTS) is cleaved; however, defects in mitochondrial import of ATFS-1 lead to trafficking of the transcription factor into the nucleus [182]. Within the nucleus, ATFS-1 activates transcription of genes that promote mitochondrial proteostasis genes and OXPHOS complex assembly [183]. Additionally, mitochondrial accumulated ATFS-1 binds to the mtDNA and limits mtDNA transcription of mitochondrial-encoded mRNAs, ultimately coordinating biogenesis and the proteostasis capacity [183]. General Control Nonderepressible 2 (GCN2) is the eIF2α kinase responsible for responding to amino acid depletion and oxidative stress [184]. It acts in a complementary pathway to ATS-1, where ATS-1 regulates mitochondrial chaperones while GCN-2 phosphorylation of eIF2α and subsequent activation of activating transcription factor 4 (ATF4) leads to attenuation of global translation [184]. In mammals, ATF5 is regulated and acts similarly to ATFS-1 [185]. It localizes to the mitochondria in the absence of stress and traffics to the nucleus in stress conditions to activate mitochondrial protein homeostasis machinery [185].

CCAAT/enhancer-binding protein (C/EBP) homology protein (CHOP) is a transcription factor activated during mitochondrial proteotoxic stress, as well as amino acid deprivation and glucose starvation [186]. Its transcriptional targets overlap with ATF4, another major transcription factor in the ISR. ATF4 and CHOP interact to induce genes involved in protein synthesis, like the chaperones Hsp60 and Hsp10, mtDnaJ, and Clp [187]. CHOP also acts to regulate ATF4 expression to prevent excessive activation of the ISR [186]. Loss of CHOP leads to disruption of mitochondrial integrity and supply of critical metabolites [186].

Additionally, there appear to be different UPR^mt^ responses depending on the mitochondrial compartment—one that is activated by matrix stress, as described above, and a different response to intermembrane space (IMS) stress [188]. The latter is stimulated by ROS overproduction, which activates Akt and phosphorylates estrogen receptor α (ERα). In turn, ERα leads to an increase in NRF1 transcription and ultimately biogenesis [189]. This response elevates levels of the proteasome and OMI, a protein that is essential for protein quality control and limiting IMS stress [188].

Interestingly, induction of ATF4, although activated during mitochondrial translation inhibition, does not lead to UPR^mt^, but simply the ISR gene expression [190]. UPRmt remains not well understood, with several questions remaining [101].

##### Mitochondrial Dysfunction and Oxidative Stress

While other ISR pathways are becoming clear, mitochondrial dysfunction and oxidative stress appear to stimulate the ISR in different ways, depending on cell type and physiological conditions [191]. Mitochondrial dysfunction and oxidative stress often go hand in hand, making it difficult to discuss one without the other.

In five conditional knockouts of mtDNA expression and maintenance genes (Twinkle, Tfam, Polrmt, Lrpprc, and Mterf4), mice suffered from mitochondrial cardiomyopathy, changes in mtDNA copy number, and decreases in respiratory activity [192]. A transcriptomic and proteomic study revealed that eIF2α signaling was enriched, specifically with ATF4, Myc, CHOP, and ATF5 upregulated [192]. In addition to ISR activation, mitochondrial one-carbon metabolism was upregulated, and this remodeling occurs early in the stress response [192]. A secondary deficiency in coenzyme Q (CoQ) develops and progresses through the disease state [192].

Many labs have observed the cytokine fibroblast growth factor 21 (Fgf21) upregulated during the stress response and have indicated it as a potential disease biomarker [191,193]. As a key regulator of lipid and glucose metabolism, it is not surprising that Fgf21 may play a role in the progression of the ISR. Forsström et al. described three temporal stages of ISR where initial respiratory deficiency modulates Fgf21 expression as well as ATF5 and remodeling of the one-carbon cycle [191]. The second stage is Fgf21-dependent and is characterized by the activation of de novo serine biosynthesis, glucose uptake, and transsulfuration, as well as upregulated expression of ATF4 [191]. Serine is a major source of one-carbon units in folate metabolism, and its upregulation via ATF4 induction may act to maintain cellular one-carbon availability following metabolic remodeling [194]. The final stage, a mild UPR^mt^, is independent of Fgf21 [191].

Inhibition of respiratory complexes can activate different branches of the ISR depending on the cell type and the affected complex [178]. In myoblasts, inhibition of complex I and III, by piericidin and antimycin, respectively, led to an increase in the NADH/NAD+ ratio in both the mitochondrial and the cytosol, hindering aspartate synthesis and depleting asparagine [178]. This activates the ISR via the eIF2α kinase GCN2, which primarily senses amino acid deficiency [178]. ATF4 was the most enriched protein, upregulating cytosolic tRNA synthetases and translation factors, amino acid transport, and biosynthesis genes [178]. Gene enrichment analysis also identified downregulation of cell proliferation, cell cycle, DNA replication, and cholesterol biosynthesis [178]. This response was not seen in myotubes [178]. In contrast, inhibition of ATP synthetase activated a strong ISR in myotubes and not myoblasts, and through a completely distinct mechanism, more related to IMM hyperpolarization [178]. Another study used CCCP, a potent uncoupler of the respiratory chain, to activate the ISR in HepG2 cells [195]. In this case, the ISR was mediated by HRI and had crosstalk with the UPR, mTORC1 activation of AMPK, and autophagy [195]. Separate studies by Fessler and Guo have better defined this pathway [196,197]. CCCP-mediated stress disrupts the mitochondrial membrane potential and triggers the activation of the mitochondrial protease OMA1 [196]. OMA1 cleaves DELE1, a poorly characterized mitochondrial protein thought to play a role in apoptosis [197]. The shortened form of DELE1 then accumulates in the cytosol, where it binds and activates HRI and initiates the ISR [196,197]. ATF4 and CHOP induction follows [195,196,197].

Altered mitochondrial dynamics have also been seen to activate the ISR via the ER stress branch led by PERK. In separate studies, Mfn2 and Drp1 were ablated [193,198]. Because ER-MAMs are intimately involved in mitochondrial dynamics, ablation of the related proteins affects both organelles [193,198]. Ablation of Mfn2 resulted in swollen mitochondria, enhanced ROS production and calcium overload, and reduced respiration [198]. Silencing PERK restored proper mitochondrial function and morphology [198]. Muñoz et al. propose the Mfn2 negatively regulated PERK via physical contact as loss of Mfn2 enhanced PERK phosphorylation and the ISR response [198]. In a Drp1 forebrain neuron-specific knockout, ATF4 was found to be upregulated, inducing the expression of Fgf21 in the brain [193]. Both authors propose that the ER stress also activated additional branches of the stress response, including pathways stimulated by impaired amino acid metabolism and heme biosynthesis, and the UPR^er^ [193,198]. Hindered mitophagy, via Miro1 knockout, upregulates mitofusins and produces a hyperfused mitochondrial network [199]. Although it is unclear which kinase is responsible in this model, it results in hyperactivation of ISR, which can have pathological consequences, including implications in neurodegeneration [193,199,200].

#### 3.2.8. Heat Shock Response

Although the name implies a response solely to temperature, the heat shock response (HSR) is induced by other stresses, including oxidative stress and heavy metals [201]. Following stress, heat shock transcription factor 1 (HSF1) activates transcription of heat shock proteins (HSPs) that act as chaperones to refold and clear accumulated misfolded proteins [201]. Agarwal et al. found that cells exposed to 42 °C for one hour showed increased mitochondrial localization in the perinuclear region [201]. This clustering resolved in the recovery hours following [201]. They suggest that increased proximity to the nucleus allows for augmented ROS levels in the nucleus, activating HIF-1α, as described in Al-Mehdi et al. [23]. However, in this pathway, HIF-1α induces HSF1 activation, leading to the HSR [201].

Heat shock also increases the expression of mitochondrial single-strand DNA-binding protein 1 (SSBP1), which forms a complex with HSF1 [202]. SSBP1 is involved in the replication and maintenance of mtDNA, but upon heat shock, it translocates to the nucleus [202]. This translocation is triggered via heat shock-induced PTP opening and aided by cytoplasmic HSF1 [202]. SSBP1-HSF1 complex upregulates chaperone expression and protects cells from proteotoxic stress during heat shock [202]. Downregulation of SSBP1 leads to decreased mtDNA copy number and activates calcineurin-mediated pathway [203].

#### 3.2.9. Mitochondrial Metabolism and Epigenetic Modifications

Epigenetics refers to chemical modifications on DNA or histones that affect their expression. More often, these include histone acetylation, deacetylation, and methylation, as well as direct DNA methylation. Histone acetylation is associated with euchromatin, while deacetylation is associated with heterochromatin, corresponding with transcription and gene repression, respectively. Histone methylation creates a docking site for chromatin-associated proteins that can then recruit other chromatin-modifying proteins. The effect of histone methylation on gene expression varies based on the number of methyl groups added. Finally, DNA methylation is a modification to the DNA itself. It is a more stable modification but can change during embryogenesis and aging. Interestingly mtDNA can be methylated; however, the literature is significantly lacking [204,205].

While nuclear epigenetic modifications can affect mitochondrial function, mitochondrial function and substrates can also affect the nuclear epigenome. In particular, TCA cycle intermediates have been shown to influence cellular physiology through epigenetic modifications, which we will discuss below [125].

##### Acetyl-CoA

Acetyl-CoA is a metabolite intermediate produced in the TCA cycle and used further in the TCA cycle, as well as the synthesis of fatty acids and sterols [125,130]. Additionally, Acetyl-CoA is utilized for histone acetylation and, therefore, gene activation [130]. High levels of Acetyl-CoA lead to increased histone acetylation, promoting cell growth and proliferation [125]. In fasted states, acetyl-CoA is generated or transported into the mitochondria for ATP synthesis [130]. This will result in lower pools of acetyl-CoA in the cytosol and nucleus, limiting fatty acid synthesis and histone acetylation [130]. Depletion of mtDNA diminished TCA cycle activity and, therefore, the available pool of acetyl-CoA, decreasing acetylation of specific histone H3 marks [206].

##### α-Ketoglutarate

In addition to being a key intermediate of the TCA cycle, FE (α-KG) is a co-substrate for 2-oxoglutarate-dependent dioxygenases (2-OGDD), a superfamily of enzymes involved in many biological processes, including the hypoxic response and chromatin modifications [125]. As a substrate of some chromatin-modifying enzymes, the availability of α-KG influences gene expression by regulating histones and DNA demethylases [125]. It has an especially important role in macrophages, promoting an anti-inflammatory pathway via histone modification while repressing the pro-inflammatory response via the NK-kB pathway [125]. 2-hydroxyglutarate (2-HG), a metabolite derived from α-KG, will be discussed later regarding its implications in carcinogenesis.

##### Succinate and Fumarate

Succinate is a TCA cycle metabolite and a product of 2-OGDD reactions [125]. Accumulation of succinate inhibits 2-OGDDs, and through this feedback, its availability influences histone and DNA methylation [125]. Succinate-mediated inhibition of 2-OGDD allows for the stabilization of HIF-1α and thereby influencing metabolic gene expression [125]. It has also been shown to induce the transcription of cytokines in activated macrophages [125]. Fumarate is a TCA cycle metabolite formed by the oxidation of succinate. Fumarate is also able to inhibit 2-OGDDS, therefore acting as a HIF-1α stabilizer [125]. It can also act as an immunomodulator by inducing histone modifications [131].

##### NAD+/NADH

Nicotinamide adenine dinucleotide is a key electron carrier in the electron transport chain. The ratio of NAD+ to NADH is an indicator of metabolic status, typically maintained 100:1 [127]. NAD+ is a co-enzyme and co-substrate for many NAD+-dependent enzymes, including sirtuins and PARPs [126]. Both sirtuins and PARPs use NAD+ as a co-substrate in DNA repair pathways and histone modifications [126]. As a cofactor for sirtuin, NAD+ levels influence the activation of PGC1α through deacetylation and downstream transcriptional responses, such as mitogenesis and fatty acid oxidation [125,127]. Circadian oscillations of NAD+ have also been observed [128,129]. Through these fluctuations, NAD+ is able to activate nuclear sirtuins and further influence daily cycles of energy storage and utilization [128]. PARPs use NAD+ in the ADP-ribosylation of histones, marking it for DNA repair, and have associations with DNA modifications [126]. A deficiency of NAD+ can promote DNA methylation through this and other pathways [126].

##### FAD

Flavin adenine dinucleotide is a metabolite derived from the vitamin riboflavin. It is produced in the mitochondria and acts as an electron carrier. FAD also acts as a cofactor for lysine demethylases [126]. The lysine demethylase LSD1, in particular, regulates mitochondrial respiration and energy expenditure, and therefore alterations in the FAD/FADH ratio, which fluctuates with other metabolic activities like fatty acid oxidation and the TCA cycle, can affect LSD1-mediated demethylation [126,132].

##### mtDNA

Although not a metabolite, it is interesting to note that mtDNA depletion was found to alter nuclear genome methylation patterns in cancer genomes [207,208]. The mitochondrial mass itself influences mRNA abundance, translation through chromatin modifications, and alternative splicing [209].

#### 3.2.10. Other Players

G-Protein Pathway Suppressor 2 (GPS2) is a regulator of inflammation and lipid metabolism. It is a dual localizing protein, where its nuclear presence plays roles as corepressor and coactivator of several transcription factors. Cardamone et al. identified it as a mediator in retrograde signaling [102]. Upon depolarization of the mitochondria membrane, GPS2 translocates from the mitochondria to the nucleus and activates transcription of nuclear-encoded mitochondrial genes and stress-response genes [102]. Loss of GPS2 promotes tumor growth through the activation of AKT signaling [210].

Other dual localizing proteins include COQ7, PDC, and Nrf2. Coenzyme Q7, Hydroxylase (COQ7) in the mitochondria is involved with the biosynthesis of ubiquinone. In response to high levels of ROS, COQ7 is trafficked to the nucleus, where it associates with chromatin to regulate cellular ROS levels by promoting glutamine metabolism and suppressing a subset of UPR^mt^ genes [211]. Pyruvate dehydrogenase complex (PDC) is an enzyme that converts pyruvate to Acetyl-CoA. In addition to this role, PDC links mitochondrial metabolism with nuclear gene expression to regulate cell growth. In response to impaired oxidative phosphorylation, PDC was reported to translocate from the matrix to the nucleus to generate pools of Acetyl-CoA for histone acetylation [212]. PDC could also translocate in response to growth signals [212]. Nuclear factor erythroid 2-related factor 2 (Nrf2) is localized on the outer mitochondrial membrane in a complex with KEAP1 and PGAM5 [213]. Increased levels of ROS triggers NRF2 to localize in the nucleus to activate antioxidant defenses [212]. Additionally, decreased mTOR activity via long-term exposure to rapamycin increases the turnover of autophagy adaptor p62/SQSTM1, displacing KEAP1 from Nrf2 [214]. An increase in nuclear Nrf2 results in upregulation of NRF1 and TFAM and, therefore, mitogenesis [214].

Mitochondrial-derived vesicles have been identified as intracellular transporters between mitochondria and peroxisomes, but it is possible that the mitochondria could communicate with other organelles, such as the nucleus, in this manner as well [215,216].

## 4. Pathological Consequences

Mitochondrial diseases are a group of genetic diseases caused by mutations in nuclear DNA or mtDNA encoding for mitochondrial proteins and RNAs, especially tRNAs [217,218]. The mitochondrial genome has a mutation rate 10–17 fold higher than nuclear DNA, and over 250 pathogenic mutations have been identified [219]. Primarily, the mutations affect OXPHOS [217]. These diseases are often multisystemic and mostly affect tissues with high energy demand, such as muscle and the brain [217,220]. While mitochondrial diseases are hereditary, they have complex genetics—nuclear mutations can be inherited in an X-linked, dominant, or recessive fashion, whereas mtDNA mutations are maternally inherited [217]. Age-related accumulation of mutations also may contribute to mitochondrial dysfunction [221]. This becomes more complicated with the heteroplasmic nature of mtDNA. Because each cell contains thousands of copies of mtDNA, there may be a mixed population of wild-type and mutated mtDNA [222]. If heteroplasmy favors wild-type mtDNA, there will be little to no clinical signs of mitochondrial disease. When the mutated mtDNA levels hit a biochemical threshold, symptoms will be prevalent and typically increase in severity as the percent of mutated mtDNA increases [220].

### 4.1. Defects of mtDNA Maintenance and Expression

Maintenance and expression of mtDNA heavily rely on nuclear-encoded factors, as discussed earlier in the review. Defective mtDNA maintenance or mitochondrial gene expression prevents proper assembly of the respiratory chain complexes and the defective oxidative phosphorylation [218]. The resulting pathology is tissue-specific and can range from muscle weakness to stroke-like episodes and nervous system disorders in adults [218].

Like nuclear DNA, mtDNA has its own repair pathways. The main repair pathway is base excision repair (BER), and it is largely repairing ROS-induced lesions [223,224]. Other repair pathways, such as mismatch repair, translesion synthesis, and single- and double-strand break repairs, are less efficient or have not been shown to occur in mitochondria [224,225].

### 4.2. Mitochondrial Function in Carcinogenesis

Cancer cells illustrate how mitochondrial function can affect the nuclear control of cell growth. Cancer cells develop mutations and adapt to grow uncontrollably, and spread throughout the body. One of these adaptions is metabolic reprogramming, known as the Warburg Effect. The major concept of the Warburg Effect is an increase in glucose uptake and lactate production in cancerous cells. While this can be true, mitochondria do not lose the ability to create ATP via oxidative phosphorylation in cancer cells; in fact, it would not be beneficial for the cells to have defective mitochondria. Metabolic reprogramming is an essential step in tumor proliferation.

As discussed earlier, mitochondrial metabolites play a role in epigenetic expression. This is also the case in tumorigenesis, with the cell using mutations to its advantage. Mutations in the mitochondrial metabolic enzymes, isocitrate dehydrogenase (IDH1/2), succinate dehydrogenase (SDH), or fumarate hydratase (FH), results in an abundance of oncometabolite (D-2-HG, fumarate, and succinate, respectively) [226]. FH and SDH each undergo a loss of function, while IDH has a gain of function [226]. All three of these oncometabolites can inhibit a-ketoglutarate enzymes, which will alter the expression of genes involved in cell differentiation and malignant transition [226]. Hypermethylation is a common epigenetic modification in cancer cells [226]. Through these modifications, further dysfunction of mitochondria may occur, increasing the production of ROS thought to aid in tumor progression [226]. Oncometabolites can also induce a pseudohypoxic response [125]. D-2-HG and succinate both act as inhibitors to PHDs, allowing HIFs to accumulate in the nucleus and promote the hypoxic response [125]. Succinylation, a post-translational modification by fumarate, can impair protein function and thereby alter downstream signaling [226]. Uncoupling proteins (UCPs) are proteins that accumulate in the IMM and alter the mitochondrial membrane potential to metabolically shift to another source of energy [227]. It was first shown during cold acclimation when the cells utilized fatty acid oxidation instead [227]. UCP2 appears to be overexpressed in many chemo-resistant cancers [227]. High rates of glycolysis not only supply energy for the cell but also replenish intermediates required for mitochondrial fatty acid oxidation [227]. One of these intermediates is Acetyl-CoA, which can elevate histone acetylation and promote cell growth and proliferation in cancer cells [125]. Tumor cells are able to interchangeably use energy sources in response to their fluctuating microenvironment [228].

As described earlier in Biswas et al., depletion of mtDNA leads to calcium dysregulation and retrograde signaling. Mitochondrial DNA deletions and depletions are common in cancers, making this pathway relevant to carcinogenesis [229]. In human pulmonary carcinoma A549 cells, Amuthan et al. observed that calcium dysregulation activated two major pathways: Ca^2+^-Calcineurin and Ca^2+^-PKC [105,230]. The former pathway regulates the activation and translocation of nuclear transcription factors [105]. Alteration of this pathway led to an increase in anti-apoptotic markers and consequently resistance to apoptosis [105]. They also observed an increase in invasive behavior that they believed to be associated with Ca^2+^-PKC pathway [230]. Another notable change in these cells was induction of glycolysis and gluconeogenesis [105]. Guha et al. further found that the mtDNA stress-activated calcineurin led to increased activity of insulin-like growth factor-1 receptor and increased glucose uptake, supporting the idea of a metabolic shift in tumor proliferation [231]. The change is mediated by Akt1, an AKT Serine/threonine kinase, which mediates transcription activation via phosphorylation of transcriptional coactivators, in this case, those involved in the mitochondrial stress response [232]. One of those co-activators is HnRNPA2 which regulates oncogenes via alternate splicing and modulates many genes involved in tumor metabolism [109,111,233].

Depletion of mtDNA has also been associated with epithelial-mesenchymal transition (EMT) [234]. Using MCF7, a malignant mammary cell line, Guha et al. induced mitochondrial stress and dysfunction by depleting mtDNA copy numbers [235]. As previously described, this activated Ca^2+^-Calcineurin signaling and induced pathways involved with EMT [235]. These cells have increased migratory capacity or invasive behavior due to an increase in transcription factors Snail, Slug, and Twist, which transcriptionally represses the epithelial cell adhesion marker cadherin, and an increase in matrix mettaloprotease, which aids in the breakdown and removal of the extracellular matrix [235].

In general, persistent oxidative stress leads to DNA damage, which can accelerate cancer growth and metastasis [236]. The majority of sequenced cancers harbor some mtDNA mutations [236]. Surprisingly, many of these mutations do not result in an apparent phenotype; however, they may be involved in increased ROS production [236]. ROS not only can further damage DNA but stimulates signaling cascades that promote cell proliferation, growth, and resistance to apoptosis [228,236]. For example, defects in Complex I lead to an accumulation of ROS, which activates Akt [237]. An increase in Akt signaling results in an upregulation of HIF-1α and anti-apoptotic proteins [237]. The Akt signaling pathway is also stimulated under hypoxia. Cancer cells, especially those in the core of the tumor, can undergo hypoxic conditions as they outpace their oxygen supply. In hypoxic conditions, Akt is recruited to the mitochondria at a higher rate where it phosphorylates pyruvate dehydrogenase kinase 1 (PDK1) [238]. The Akt-PDK1 axis is important for metabolic programming and tumor cell proliferation, and increased activity has been associated with a poor prognosis [238].

Mitochondria have become a promising focus for cancer therapeutics. Both mtDNA and mitochondrial stress signaling pathways pose as targets for therapeutic intervention [239,240]. In particular, it may be beneficial to target Akt, HnRNRA2, calcineurin, and even the UPR^mt^ as an anti-cancer strategy [233,239,240]. Mutations in mtDNA could be approached via mitochondrial transplantation—increasing the population of healthy mitochondria—allotopic gene expression (expression of a mitochondrially encoded gene from nucleus transfected constructs), or even mtDNA editing enzymes [239,241,242,243,244]. For a more in-depth look at specific mtDNA mutations and their role in cancer, the following literature provides a comprehensive view (Chatterjee et al. 2006, Girolimetti et al. 2020) [243,245].

### 4.3. Nuclear-Mitochondrial Dysfunction in Neurological Disorders

Mitochondrial diseases disproportionately affect tissues with high energy demands, meaning the nervous system is at particularly high risk. Primary mitochondrial diseases affecting the nervous system include Leigh syndrome, Alpers–Huttenlocher syndrome, Leber’s hereditary optic neuropathy (LHON), mitochondrial encephalopathy, lactic acidosis, and stroke-like episodes (MELAS). Impaired mitochondrial dynamics are also implied in the phenotype of several neurodevelopmental disorders and possibly also neuropsychiatric disorders such as schizophrenia [246]. In addition to the nervous system being affected in primary mitochondrial diseases, dysfunction mitochondria are also thought to play a role in other neurological and adult-onset neurodegenerative diseases, such as Parkinson’s and Alzheimer’s diseases. Below we will briefly describe how mitochondrial dysfunction and retrograde signaling play a role in neurodegenerative diseases.

#### 4.3.1. Parkinson’s Disease

Parkinson’s disease (PD) is a progressive nervous system disorder that affects movement, characterized by tremors and impaired posture and balance. The disease can be genetic or environmentally triggered; however, both are distinguished by the death of dopaminergic neurons in the basal ganglia. Several mitochondria-associated genes have been seen to be mutated in rare juvenile-onset PD, including mitophagy genes PINK1 and Parkin and the chaperone DJ-1 [246,247]. Mitophagy is not discussed at length in this review; however, it is a key component of mitochondrial quality control. Recently, TSPO, described earlier as an aid in nuclear-mitochondrial contacts, was found to be upregulated in PD [228]. Frison et al. used neurotoxins, rotenone, 6-OHDA, and MPP+ increased ROS in SH-SY5Y cells. ROS-activated ERK phosphorylation led to an increase in TSPO, which can then prevent mitochondrial ubiquitination and impair mitophagy [248,249]. An accumulation of defective mitochondria can result in chronic activation of retrograde signaling pathways and damage-inducing mtROS production.

Complex I deficiency appears to be a major factor in PD pathogenesis [247]. A 2011 study showed a significant association of PD with the use of pesticides classified as complex I inhibitors, such as rotenone [250]. Even though rotenone is thought to have a short half-life, research suggests that short-term exposure can induce PD-like effects in rodents later in life [250]. Oxidative stress is through to play a role in PD pathogenesis, with mitochondria being both the main source and a target [251]. mtROS induced damage of the mitochondria enhances ROS production, further contributing to neuronal cell death [251].

HtrA2, a serine protease in the intermembrane space, is thought to be a component of the stress sensing pathway with PINK1 [252]. Knockout of HtrA2 increased sensitivity to mitochondrial stress, enhanced apoptosis, and upregulated CHOP expression [252]. Increased CHOP expression has previously been associated with neurodegeneration triggered by ischemia and Charcot Marie Tooth 1B neuropathy, as well as neurotoxin models of parkinsonism [252,253]. HtrA2 KO affects the correct folding of inner membrane proteins, including respiratory complexes, leading to an increase in damaging ROS and accumulation of proteins [252]. Surprisingly, this does not activate CHOP-mediated UPR^mt^, but rather the ISR, which can be detrimental when chronically activated [246,252]. Antioxidant treatment of HtrA2 KO mice suppressed neurodegeneration and decreased their akinetic phenotype, indicating antioxidant therapy as a potential PD treatment [251,252].

#### 4.3.2. Alzheimer’s Disease

Alzheimer’s disease (AD) is a progressive neurodegenerative disease characterized by memory deficits and cognitive decline. The major hallmarks are the accumulation of beta-amyloid (Aβ) plaques and neurofibrillary tangles. Regional hypometabolism is common in AD, initially suggesting deficits in mitochondrial function [254]. Indeed, complex I deficiencies are consistently observed in AD patients [255]. In addition to OXPHOS dysfunction, increased oxidative stress and decreased antioxidant defenses are observed [255]. AD mitochondria have reduced mitogenesis, as well as impaired mitophagy and increased fission, leading to decreased mtDNA copy numbers while allowing for damaged organelles to accumulate [246,255,256]. Proteostasis is also impaired, triggering UPR^mt^ and ISR transcriptional responses [246,255].

Dysregulation of mitochondrial calcium homeostasis has long been implicated in AD pathology. Cellular calcium concentration must be highly regulated for proper cellular functions. In neurons, Ca^2+^ plays different roles depending on spatial localization and neuronal type [257]. Regulation of Ca^2+^ is important for both synaptic transmission and vesicle recycling [258]. Calcium overload, as induced by prolonged stimulation of glutamate receptors, consequently results in cell death, a process termed excitotoxicity [18]. In AD, impaired homeostasis may be due to increased ER-MAM contacts or altered expression of mitochondrial ion exchangers, such as NCLX (Na^+^/Ca^2+^ exchanger) [259]. An increase in mitochondrial Ca^2+^ can stimulate ROS production and decrease ATP production, both of which can activate retrograde signaling pathways, as well as provoke PTP opening and induce apoptosis [259]. AD brains have an increase in Aβ-containing mitochondria, which can augment respiratory deficiency and further enhance ROS production [247,254]. While mitochondrial dysfunction is common in AD and likely plays a role in disease progression, some authors suggest mitochondrial deficits as the primary insult in sporadic AD pathology, known as the mitochondrial cascade hypothesis.

#### 4.3.3. Amyotrophic Lateral Sclerosis (ALS)

Amyotrophic lateral sclerosis (ALS) is a progressive neurodegenerative disease characterized by motor neuron degeneration in the brain and spinal cord. Approximately 90% of cases are sporadic (sALS), while only 10% are familial (fALS). A few of the major mutations linking mitochondrial dysfunction and ALS are mutations in SOD1, TDP-42, and FUS [246]. These mutations activate the UPR^mt^, and may also play a role in pro-apoptotic signaling [246,260]. Although there are differences in mitochondrial characteristics between the types, both display decreased mitochondrial membrane potentials, altered mitochondrial morphology, and decreased respiratory activity [261,262]. Mitochondria appear swollen with a fragmented network [260]. Delic et al. recorded abnormal mitochondrial distribution and density in sALS motor neurons, which could affect signaling between the mitochondria and the nucleus [262]. Both types also showed increased intracellular calcium levels [261]. Dysfunctional calcium regulation, as we have discussed, can activate a variety of retrograde signaling pathways. Increased ROS and ROS-associated damage have been observed in ALS cell lines [260].

## 5. Conclusions and Perspectives

Mitochondria act as signaling hubs in the cell, interacting with other organelles through signaling pathways and direct contact sites. The relationship between the mitochondria and the nucleus is critical for cell survival, influencing energy production, metabolism, cell proliferation, and more. Signaling from the mitochondria to the nucleus, or retrograde signaling is facilitated by a variety of molecules and pathways. Calcium and ROS have been described most in detail, but newer literature has shown roles for ncRNAs, metabolites, and mtDNA itself.

As mitochondrial dysfunction heavily affects tissues and cells with high energy demand, it can be expected that disruption of retrograde signaling would disproportionately affect the same tissues. However, there are many different pathways and outcomes of retrograde signaling. Retrograde signaling is best characterized in proliferative cells; however, it is thought to play a role in all cell types, albeit not ubiquitous. Further research is needed to discern the pathways present in each cell type and how they are altered in disease models. By understanding the major molecules and proteins involved, therapeutics can be designed to target the activated pathways and curtail disease phenotypes.

Having established that mitochondria share both functional and physical interactions with other organelles, it is not irrational to hypothesize, at the very least, close contact between the mitochondria and the nucleus, potentially facilitating signaling. Some literature has described an increase in proximity between the mitochondrial and the nucleus under various conditions; however, it has yet to be concretely described or microscopically captured that the two organelles share direct contacts. Through reporter assays, transcriptomics, and super-resolution imaging, we can not only analyze conditions that stimulate mitochondrial re-localization to the perinuclear region but better understand how mitochondrial distribution affects retrograde signaling and potentially identify docking proteins in cases of close proximity.

## Figures and Tables

**Figure 1 biomolecules-12-00427-f001:**
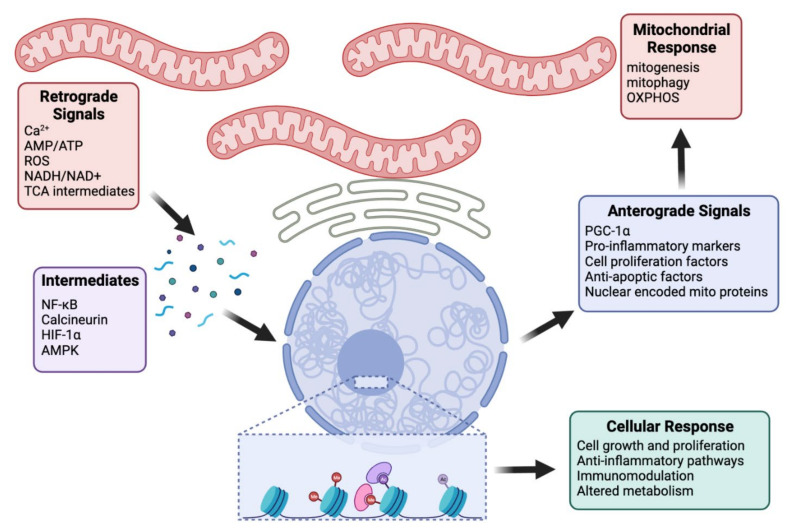
Signaling from the mitochondria to the nucleus. Signals originating from the mitochondria, including calcium, reactive oxygen species (ROS), and AMP/ATP, often stimulate pathways, leading to transcriptional changes in the nucleus. Nuclear responses can involve upregulating cell proliferation and anti-apoptotic factors, as well as proteins involved in mitogenesis, such as PGC-1α and nuclear-encoded mitochondrial proteins. The levels of other molecules, such as TCA intermediates Acetyl CoA and α-ketoglutarate, can influence the epigenome by modifying methylation and acetylation, and consequently, the cell physiology.

**Table 1 biomolecules-12-00427-t001:** Nuclear effectors of mitochondrial signaling.

Nuclear Signal	Caused by	Mediated by	Mitochondrial Response	Ref
TFAM	OXPHOS defect	PGC-1α, NRF1/2	Transcription initiation, mtDNA maintenance, and stabilization	[65,66]
PARP1	Nuclear DNA damage		Decreased metabolism and mitogenesis, increased oxidative stress	[69]
ERRα	exercise	PGC-1α, mTOR, cAMP	Oxidative metabolism, metabolism remodeling	[70,71]
CREB	Low ROS levels, DFO	Mitochondrial PKA	Expression of ETC components	[70,72,73]
SOD2	Accumulation of ROS and free radicals	?	ROS degradation	[74]
MEF2D	Phosphorylation by CaMK	Hsp70	Complex I function	[75,76]
NF-κB	TNFα stimulation	IkB, Hsp70, p53	Decrease mitochondrial gene expression	[77,78]
TERT	Oxidative stress	Src kinase, Ran GTPase	mtDNA protection	[79,80]
			Reverse transcription of mitochondrial tRNAs	[81]
STAT3			Modulation of ETC	[82]
P53	Pro-apoptotic stimuli, oxidative stress	Tid1	Apoptosis, necrosis	[83,84,85]
	Oxidative stress		Reduces SOD2 scavenging capacity	[86]
		POLG	mtDNA stability, replication, and repair	[87,88,89]

**Table 2 biomolecules-12-00427-t002:** Mitochondrial signals that trigger a nuclear response.

Mitochondrial Signal	Caused by	Mediated by	Nuclear Response	Ref
Calcium	mtDNA depletion, ΔΨm	NF-κB, JNK, ATF2, calcineurin, NFAT	Ca ^2+^ homeostasis Glucose metabolism Pro-inflammatory factors Cell proliferation factors Anti-apoptotic factors	[103,104,105,106,107,108,109,110,111]
ROS	Hypoxia, defects in mitochondrial respiration	HIF-1α	Hypoxic transcriptional response	[23,112,113]
NO	Calcium	cGMP, PGC-1α	Mitogenesis	[114,115,116,117]
AMP/ATP	Cellular stress, fasting, exercise	AMPK	Mitogenesis, mitophagy	[118,119,120,121,122,123,124]
NADH/NAD+	Metabolic activities	SIRTs, PGC-1α, PARP	Mitogenesis, fatty acid oxidation, DNA repair, DNA modifications	[125,126,127,128,129]
Acetyl CoA	Fed states	acetyltransferases	Histone acetylation, cell growth, and proliferation	[125,130]
α-ketoglutarate	TCA cycle	2-OGDDs	Hypoxic response, chromatin modifications	[125]
Succinate	TCA cycle	HIF-1α	Histone and DNA methylation	[125]
fumarate	Oxidation of succinate	HIF-1α	Histone modifications	[125,131]
FAD/FADH	Metabolic activities		demethylation	[126,132]

## Data Availability

Not applicable.

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
