# Peer review of "Nuclear-Mitochondrial Interactions"

_biomolecules, 2022, doi:10.3390/biom12030427_

Round 1

Reviewer 1 Report

This manuscript Nuclear-Mitochondrial Interactions” by Walker and Moraes described the signaling pathway between the mitochondria and nucleus. Authors have explained the various ways of interaction of mitochondria with the nucleus.

I appreciate the author and co-author for writing a well-organized research paper. Author should read the review article thoroughly and set its style as per journal format.

Suggestion!

At the line number 394, authors clearly explained the role of Telomerase RNA (TERC-53) in cell senescence and at the line number 632, you have mentioned about the mutagenic rate in mitochondria. Keeping in view,

You know that

*Telomer shortening occurs due to environmental and genetic factor.

If the literature is available, could you please add at least one paragraph/sub-heading about the effect of environmental factors, telomere shortening and mutagenesis/changes in mitochondria.

It’s a suggestion, authors can disagree.

Author Response

Reviewer 1

This manuscript Nuclear-Mitochondrial Interactions” by Walker and Moraes described the signaling pathway between the mitochondria and nucleus. Authors have explained the various ways of interaction of mitochondria with the nucleus.

I appreciate the author and co-author for writing a well-organized research paper. Author should read the review article thoroughly and set its style as per journal format.

Suggestion! At the line number 394, authors clearly explained the role of Telomerase RNA (TERC-53) in cell senescence and at the line number 632, you have mentioned about the mutagenic rate in mitochondria. Keeping in view, You know that *Telomer shortening occurs due to environmental and genetic factor.If the literature is available, could you please add at least one paragraph/sub-heading about the effect of environmental factors, telomere shortening and mutagenesis/changes in mitochondria.It’s a suggestion, authors can disagree.

Although we agree with the Reviewer that this is an interesting topic, we believe that this discussion would steers off the Review’s main theme, that is nuclear-mitochondrial interactions.

Reviewer 2 Report

Walker and Moraes present an inspiring and comprehensive overview of mitochondrial function with specific focus on interactions between mitochondria and the nucleus. Mitochondrial signaling is a rapidly expanding field, thus the manuscript is absolutely timely. Especially useful is the table overview of the addressed aspects coupled with the very extensive catalogue of literature. The only shortcoming of the manuscript is the lack of implications of the described pathways in human pathology (although this aspect is emphasized in the Abstract). The sections 'Pathological Consequences' and 'Defects of mtDNA maintenance and expression' contain only very general statements on mitochondrial diseases of genetic origin and lack a connection with the main topic of the manuscript: mitochondrial signaling. These sections should either be better integrated into the manuscript or just removed. Apart from this, I have only minor formal remarks and recommend the paper for publication.

1. Section 'Pathological Consequences': pathogenic mtDNA mutations occur not only in protein-coding parts of the mitochondrial genome, but also in ribosomal and transfer RNA genes. The most frequent pathogenic mutations are actually tRNA mutations. (Line 631)
2. Section 'Nuclear control': not all nuclear-encoded mitochondrial proteins carry an N-terminal mitochondrial targeting sequence (MTS). A group of inner-membrane proteins have internal (cryptic) targeting sequences and are imported by another translocation pathway (TOM70/TOM22 and TIM22/TIM54). (Lines 185-191)
3. Section 'FAD': It is unclear how lines 598-601 relate to this section. 'Mitochondrial content itself influences mtRNA abundance' (lines 599-600): what is meant under 'mitochondrial content'? Possibly mtDNA copy number?
4. Section 'Other players': PDC is formally not part of the TCA cycle (lines 614-615).
5. The title of the first section seems to be incomplete. 'Mitochondria: Form and Function' or 'Mitochondrial Form and Function'? (Line 19)
6. It is difficult to distinguish different levels of sections in the manuscript. As an example, the section in lines 536-548 describes general principles, of which specific aspects are discussed in the following sections. However, all heads are formatted the same. Similar case in section 'Retrograde Signaling'.
7. Line 28: "most of the respiratory complexes". Which respiratory complexes are not in IMM? Please rephrase.
8. The abbreviation 'PTP' is not explained in the text (line 96).
9. Typos and formatting issues. Line 405: 'There five types'. Lines 140 and 728: reference number formatting. Lines 880-884: capital letters in references 55 and 56. Line 266: incomplete protein name 'HIFα'. Line 392: 'This is turn...' Line 284: 'fasting or and exercise'. Line 345: 'produced in mitochondrial,'. Line 675: 'Oncometabolites can also a pseudohypoxia response.'

Author Response

Reviewer 2

Walker and Moraes present an inspiring and comprehensive overview of mitochondrial function with specific focus on interactions between mitochondria and the nucleus. Mitochondrial signaling is a rapidly expanding field, thus the manuscript is absolutely timely. Especially useful is the table overview of the addressed aspects coupled with the very extensive catalogue of literature. The only shortcoming of the manuscript is the lack of implications of the described pathways in human pathology (although this aspect is emphasized in the Abstract). The sections 'Pathological Consequences' and 'Defects of mtDNA maintenance and expression' contain only very general statements on mitochondrial diseases of genetic origin and lack a connection with the main topic of the manuscript: mitochondrial signaling. These sections should either be better integrated into the manuscript or just removed. Apart from this, I have only minor formal remarks and recommend the paper for publication.

  1. Section 'Pathological Consequences': pathogenic mtDNA mutations occur not only in protein-coding parts of the mitochondrial genome, but also in ribosomal and transfer RNA genes. The most frequent pathogenic mutations are actually tRNA mutations. (Line 631). We emphasized this feature. (Page 13, line 652).
  2. Section 'Nuclear control': not all nuclear-encoded mitochondrial proteins carry an N-terminal mitochondrial targeting sequence (MTS). A group of inner-membrane proteins have internal (cryptic) targeting sequences and are imported by another translocation pathway (TOM70/TOM22 and TIM22/TIM54). (Lines 185-191). This segment was corrected and expanded. (Page 4, line 184).

  3. Section 'FAD': It is unclear how lines 598-601 relate to this section. 'Mitochondrial content itself influences mtRNA abundance' (lines 599-600): what is meant under 'mitochondrial content'? Possibly mtDNA copy number? We corrected this segment by clarifying that mitochondrial mass affects nuclear gene expression. It is under a “mtDNA” subheading now. (Page 13, line 619).

  4. Section 'Other players': PDC is formally not part of the TCA cycle (lines 614-615). Clarified. (Page 13, line 636).

  5. The title of the first section seems to be incomplete. 'Mitochondria: Form and Function' or 'Mitochondrial Form and Function'? (Line 19) . Clarified (Page 1, line 19)

6.It is difficult to distinguish different levels of sections in the manuscript. As an example, the section in lines 536-548 describes general principles, of which specific aspects are discussed in the following sections. However, all heads are formatted the same. Similar case in section 'Retrograde Signaling'.
We added a number system and italics to make headings, subheadings, and sub-subheadings clearer.

  1. Line 28: "most of the respiratory complexes". Which respiratory complexes are not in IMM? Please rephrase. Re-worded. (Page 1, line 25)
  2. The abbreviation 'PTP' is not explained in the text (line 96). Included. (Page 5, line 240)
  3. Ty pos and formatting issues.
  • Line 405: 'There five types'. (Page 9, line 426).
  • Lines 140 and 728: reference number formatting. (Page 3, line 139; Page 15, line 750)
  • Lines 880-884: capital letters in references 55 and 56. (Page 20, lines 974-977)
  • Line 266: incomplete protein name 'HIFα'. (Page 6, line 276)
  • Line 392: 'This is turn...' (Page 9, line 413).
  • Line 284: 'fasting or and exercise'. (Page 6, line 304).
  • Line 345: ‘produced in mitochondrial,’. (Page 8, line 366).
  • Line 675: 'Oncometabolites can also a pseudohypoxia response.' (Page 14, line 697)

Reviewer 3 Report

The manuscript entitled “Nuclear-mitochondrial interactions“ by Walker and Moraes highlights the role of mitochondria as signaling hubs in the cell and focuses on mitochondrial retrograde signaling that is mitochondria to nucleus communication for maintenance of cellular homeostasis. In particular, in case of mitochondrial dysfunction or cellular environmental perturbations, mitochondria transmit signals to the nucleus, which in turn up-regulate gene expression to modulate metabolism or initiate stress response. Disruption of retrograde signaling may also contribute to pathological defects. The work covers the known retrograde signaling pathways and the triggering molecules, the physical and functional interactions between mitochondria and the nucleus and the pathological aspects of a compromised cross-talk.

In general the manuscrit is well written and quite comprehensive. However, it requires major changes before publication, as reported below.

  1. The general feeling in reading the manuscript is that the main focus of the review, which is described in the abstract, is sometimes lost. At this regard, the narrative of the article could be improved by distinguishing sections and subsections; adding connections among sections; grouping some subsections in one, as for example grouping the metabolites involved in epigenetic modifications (acetyl-CoA, a-ketoglutarate, succinate and fumarate) or the NAD/NADH with FAD.
  2. The list of references is long, but not always necessary and appropriate. In certain cases it may be useful to cite a referral work in the field and the references therein. Some references in the text appear superfluous: for example refs. 1 and 2 (lines 20-21; line 55). A more appropriate reference for the role of Mitochondrial derived peptides (MDPs) in metabolism, aging and cell survival could be found (lines 326-327).
  3. The section on Retrograde signaling (lines 210-222) needs a careful revision.

Primarily, it should be mentioned that the retrograde response has been discovered and characterized in its molecular details in yeast. At this regard two reviews (PMID: 16771627; PMID: 30165482) should be cited. Furthermore, references 71 and 75 are not appropriate to describe the translocation of Rtg1,3 complex, the up-regulation of TCA cycle genes and activity: again PMID: 16771627 should be cited. The original work (PMID: 24605246) should be cited regarding the role of ATP as a major trigger of retrograde response in yeast.

  1. The introduction to the section Mitochondrial Metabolism and Epigenetic Modifications (lines 537-548) should contain a brief mention of the mitochondrial metabolites or mitochondria-associated molecules described further and related to epigenetic modifications.
  2. Lines 689-690. Mitochondrial DNA deletions and depletions and related pathways, including the retrograde signaling, are relevant to carcinogenesis. At this regard, it should be also highlighted that they represent potential targets of anticancer therapy as discussed in the work PMID 23253132.
  3. Line 706. The work PMID: 29250487 should be cited regarding mitochondrial dysfunction as a potential driver of Epithelial-to-Mesenchymal Transition (EMT) in cancer.
  4. Fig.1. The legend title should be Signaling from mitochondria to nucleus.

The legend content lacks the description of intermediates of retrograde signaling (in the box) and mitochondrial response activated by anterograde signaling (in the box). Furthermore, the TCA intermediates mentioned in the last sentence and influencing the epigenome do not appear in the figure.

  1. Tables I and II are not cited in the body text.

Minor points:

Line 140. 32 typo error?

Line 141. Eliminate J

Line 257. Eliminate “recently”.

Line 211. Change in: alterations in mitochondrial function and adaptations

Line 526. Add the reference number for Al-Mehdi et al.

Line 675. Check the sentence.  

Line 697. “The lab further found……“, change the lab with the authors‘ name

Author Response

Reviewer 3

The review entitled "Nuclear-Mitochondrial Interactions" by B.Walker et al. discusses the relationship between mitochondria and nucleus in controlling intracellular signaling, cell functions and disease pathology. Overall, the manuscript is well-written and covers the most available knowledge about mitochondria-related signaling. I only have a few suggestions to improve the quality of this manuscript.

  1. There is a growing body of studies showing the impact of intracellular calcium levels on mitochondrial function and related signaling pathways. Also, mitochondrial calcium uptake can be discussed as a mechanism for calcium homeostasis. Therefore, the authors may include a few related lines in the calcium section. We added a segment expanding on calcium. (Page 5, line 232).

  1. There is available evidence on the impact of mitochondrial-induced oxidative stress and activation of JNK pathways. Having a related paragraph in the free radical section would be very informative. We added a segment expanding on this pathway. (Page 6, line 282).

  1. The role of MAPK pathway in OXPHOS and oxidative stress control has been well established. Please include a paragraph on MAPK pathway in the metabolic sensor section. We discussed JNK and MAPK in the free radicals section as suggested. We believe that discussion covers the main concepts.

  1. MOTS-c and Small humanin-like peptides (SHLPs) should be defined in line 326. (Page 7, line 346).

  1. There are a lot of examples of the impact of mitochondrial function in carcinogenesis, so there is no need to mention examples that include non-transformed cells such as C2C12 (line 690). Removed.

  1. In line 706, the authors called MCF10A a breast cancer cell line. However, MCF10A is a non-transformed breast cell line. This mistake should be corrected.

  1. The conclusion section needs to be more comprehensive and speculative. The authors also need to give their own opinion (based on the previously discussed literature) on the discussed topic and compare the findings of different studies. We added more speculations about the physical interactions between mitochondria and nucleus. (Page 17, line 855).

Reviewer 4 Report

The review entitled "Nuclear-Mitochondrial Interactions" by R.Walker et al. discusses the relationship between mitochondria and nucleus in controlling intracellular signaling, cell functions and disease pathology. Overall, the manuscript is well-written and covers the most available knowledge about mitochondria-related signaling. I only have a few suggestions to improve the quality of this manuscript.

  1. There is a growing body of studies showing the impact of intracellular calcium levels on mitochondrial function and related signaling pathways. Also, mitochondrial calcium uptake can be discussed as a mechanism for calcium homeostasis. Therefore, the authors may include a few related lines in the calcium section.

  1. There is available evidence on the impact of mitochondrial-induced oxidative stress and activation of JNK pathways. Having a related paragraph in the free radical section would be very informative.

  1. The role of MAPK pathway in OXPHOS and oxidative stress control has been well established. Please include a paragraph on MAPK pathway in the metabolic sensor section.

  1. MOTS-c and Small humanin-like peptides (SHLPs) should be defined in line 326.

  1. There are a lot of examples of the impact of mitochondrial function in carcinogenesis, so there is no need to mention examples that include non-transformed cells such as C2C12 (line 690).

  1. In line 706, the authors called MCF10A a breast cancer cell line. However, MCF10A is a non-transformed breast cell line. This mistake should be corrected.

  1. The conclusion section needs to be more comprehensive and speculative. The authors also need to give their own opinion (based on the previously discussed literature) on the discussed topic and compare the findings of different studies.

Author Response

There were only 3 reviewers

Round 2

Reviewer 3 Report

The manuscript has been significantly improved and can now be published.